# Technology-mediated screening interviews for youth mental health: Content validation, randomized controlled trial, and expert evaluation

Izidor Mlakar[1]*, Valentino Šafran[1], Nejc Plohl[2], Hojka Gregorič Kumperščak[3], Sara Močnik[3], Žan Smogavc[3], Urška Smrke[1]

**1** Faculty of Electrical Engineering and Computer Science, University of Maribor, Maribor, Slovenia, **2** Department of Psychology, Faculty of Arts, University of Maribor, Maribor, Slovenia, **3** Child and Adolescent Psychiatry Unit, University Medical Center Maribor, Maribor, Slovenia

* izidor.mlakar@um.si

**Data availability statement:** All raw data used to generate results and to to replicate all study findings reported in the article are within the

## Abstract

Early identification of mental health disorders in children and adolescents is essential for optimal outcomes, yet access to specialist psychiatric assessment remains limited by severe workforce shortages and geographic barriers. This study examines content validity, feasibility and acceptability from youths' perspective, and clinical utility from experts' perspective of technology-mediated screening interview for youth mental health assessment through three sequential investigations (ISRCTN Registry, ISRCTN68006163, registered on 16/04/2025). First, content validity was established through expert evaluation, demonstrating that all screening questions met established psychometric criteria for relevance and clarity. Next, a randomized controlled trial evaluated feasibility and acceptability (satisfaction and willingness to repeat with the same modality) across psychiatrist-led, chatbot, and robot modalities among 106 young patients aged 10–19 receiving mental health treatment. Finally, expert clinical evaluation of recorded interviews assessed clinical utility across modalities. Psychiatrist-led interviews achieved higher satisfaction scores, particularly for communication quality, while willingness to repeat screening did not differ significantly across modalities. Personality characteristics, specifically conscientiousness and open-mindedness, predicted satisfaction with communication across all conditions. Finally, expert clinical evaluation of recorded interviews revealed comparable ratings across modalities for most dimensions, with only alignment with other assessment methods favoring psychiatrist-led approaches over robot-delivered interviews. The results show technology-mediated screening, using the proposed interview, demonstrates acceptable content validity, user feasibility from youth perspective, and clinical utility from expert perspectives. These findings support implementation of accessible, self-administered screening tools as complementary first-step assessments that

manuscript and its Supporting Information files.

**Funding:** This work has received funding by the European Union Horizon Program project SMILE (Grant number 101080923 to IM), project CERTAIN (Grant number 101189650 to HGK) and from the Slovenian Research Agency (Research Core Funding) No. P2-0069, the Young Researcher Funding (6316-3/2018-255, 603-1/2018-16 to VŠ). The funders had no role in study design, data collection and analysis, decision to publish, or preparation of the manuscript. Responsibility for the information and views expressed herein lies entirely with the authors.

**Competing interests:** The authors have declared that no competing interests exist.

maintain clinical relevance while expanding access to mental health evaluation for youth populations in underserved settings where specialist availability is limited.

### Author summary

In the research, we addressed a critical challenge in youth mental health: limited access to specialists who can conduct timely psychiatric assessments. With only one child psychiatrist available for every 350 young people needing care, many children and adolescents face significant delays in receiving help. We investigated whether technology-mediated screening interviews, conducted through tablet chatbots or humanoid robots, could expand access while maintaining clinical value. We conducted three interconnected studies assessing content validity, feasibility and acceptability, and clinical utility of the screening interview with 106 young patients aged 10–19 and mental health experts. Our findings demonstrate that while young people preferred human psychiatrists for conducting interviews, they were equally willing to repeat screenings with technology-based systems. Importantly, expert psychiatrists rated the clinical value of information gathered through technology-mediated interviews as comparable to traditional methods across most evaluation criteria. The results suggest that technology-mediated screening could serve as an accessible first step in mental health assessment, particularly in settings where specialists are unavailable. This approach has potential to improve early identification of mental health concerns while preserving the clinical rigor necessary for responsible care of vulnerable youth populations.

## 1. Introduction

Mental health disorders among children and adolescents represent a critical public health challenge, with prevalence rates reaching epidemic proportions globally [1]. Approximately 20% of adolescents experience a mental health disorder, with half of all lifetime mental health conditions emerging by age 14 [2,3]. Early identification and intervention are paramount for optimal outcomes, yet significant barriers persist in timely access to specialized psychiatric assessment [4,5]. The shortage of pediatric psychiatrists, with one specialist available for every 350 youth needing care, creates substantial delays in diagnosis and treatment initiation [6].

Traditional clinical interviews, while remaining the gold standard for mental health evaluation, are resource-intensive and may be influenced by social desirability bias, particularly among adolescents who may feel uncomfortable discussing sensitive topics with clinicians [7,8]. Studies have shown that youth may underreport symptoms or minimize severity when speaking directly with healthcare providers due to concerns about confidentiality, stigma, or fear of consequences [9,10]. Additionally, geographic barriers limit access to specialized care, with rural and underserved communities experiencing disproportionate shortages of mental health services [11].

The integration of digital technologies into mental health screening represents a promising avenue for addressing these challenges [12–16]. Recent advances in conversational artificial intelligence (AI) and social robotics have demonstrated potential for engaging young people in mental health contexts, potentially reducing barriers to disclosure while maintaining clinical utility [17,18]. Conversational agents significantly reduce symptoms of depression and distress in youth populations [19–22], while social robots have been found to create "safe spaces" for emotional sharing among children with anxiety [23]. Digital mental health interventions have shown promise in improving access and engagement among adolescents [24,25], who demonstrate high comfort levels with technology-mediated healthcare interactions. Chatbot-assisted health self-assessment approaches have demonstrated acceptability across diverse populations, with research indicating that the perceived anonymity and non-judgmental nature of chatbot interactions can facilitate disclosure of sensitive information, particularly for topics individuals find difficult to discuss with healthcare providers [26]. Furthermore, previous research has also demonstrated that children and adolescents may be more willing to disclose sensitive information to computer-based/robot systems with human likeness compared to human interviewers, potentially due to reduced perceived judgment and increased sense of privacy [27–29]. Moreover, individual personality characteristics may significantly influence user experience with technology-mediated mental health interventions. Research in digital health adoption demonstrates that conscientiousness, characterized by self-discipline and goal-directed behavior, predicts greater engagement with digital health tools and higher satisfaction with structured interventions [30–32]. Similarly, open-mindedness (openness to experience) has been consistently associated with positive attitudes toward novel technologies and willingness to engage with innovative healthcare approaches [33–35]. Furthermore, social desirability bias, may differentially impact satisfaction across modalities, with some individuals preferring the perceived anonymity of technology-based interactions [9,10,36] The potential for independent completion of mental health screening using readily available technology could transform access to early identification services, particularly in settings where psychiatric specialists are unavailable [37].

Overall, the current evidence suggests that adolescents may exhibit varying comfort levels and willingness to engage with different interview modalities, with some even preferring technology-mediated interactions for sensitive topics [38]. Despite growing enthusiasm for digital mental health screening, several critical research gaps remain unaddressed. First, while numerous studies have examined technology-mediated mental health tools in adult populations, pediatric applications remain significantly underrepresented, particularly in clinical samples with confirmed psychiatric diagnoses. Most existing research focuses on general population screening or non-clinical samples, limiting applicability to youth already engaged in mental health treatment

Despite growing enthusiasm for digital screening, direct head-to-head comparisons of modalities in clinical populations remain scarce. While available evidence suggests equivalence in measurement validity, with distinct trade-offs in feasibility and user engagement, across several formats [39,40], few have systematically compared human-led versus technology-assisted approaches within the same population or examined the content validity of screening instruments specifically designed for multi-modal delivery [24,41].

Furthermore, the clinical value of such screening approaches, as perceived by expert psychiatrist, remains largely unexplored, creating uncertainty about clinical implementation and integration into existing care pathways [42,43]. Expert evaluations of AI-assisted/ technology-mediated mental health tools reveal complex attitudes that vary across clinical utility dimensions. Recent surveys indicate that mental health professionals generally acknowledge the potential benefits of digital screening tools, with majority recognizing their value for increasing access and efficiency [44]. Moreover, studies examining psychiatrists' perceptions of AI-assisted/technology-mediated diagnostic tools show favorable ratings for efficiency and accessibility (typically above scale midpoints), but lower confidence ratings for clinical validity and integration with existing assessment protocols [45,46]. Overall, expert evaluations tend to favor human-led approaches across dimensions involving clinical judgment, therapeutic relationship quality, and alignment with established assessment frameworks, while rating technology-mediated approaches more favorably for standardization and accessibility. This pattern suggests

that while experts may provide generally positive evaluations of digital mental health screening tools, systematic preferences for complementary, but human-led approaches consistently emerge in areas that require clinical expertise and nuanced professional judgment [47].

The development of effective technology-assisted screening tools requires rigorous validation across multiple domains: content validity to ensure clinical relevance, user acceptability to confirm feasibility in target populations, and expert evaluation to establish clinical utility. Such evaluation is particularly crucial in pediatric populations, where developmental considerations, communication preferences, and varying levels of digital literacy may significantly influence the effectiveness of different assessment modalities [48,49]. To this end, the study aimed to: (1) establish content validity of screening interview questions through expert evaluation, (2) assess the children and adolescents' satisfaction with and willingness to repeat the interview in two accessible technology modalities (tablet-based chatbot and humanoid robot) compared with traditional psychiatrist-led interviews, and (3) to evaluate and determine whether the clinical value of these interviews, from a psychiatric perspective, differs across human, human-like technology (robot), and tablet-based approaches.

The central question addressed whether technology-mediated screening interviews could maintain clinical utility while enabling independent completion by youth, potentially expanding access to mental health screening beyond traditional clinical settings. Based on existing literature we propose the following set of hypotheses:

(H1) The screening interview questions will demonstrate acceptable content validity, with all questions meeting established psychometric criteria for relevance and clarity when evaluated by mental health experts

(H2) Satisfaction with the interview and satisfaction with communication will differ among the three conditions, with psychiatrist-led interviews receiving higher satisfaction ratings than technology-mediated conditions (chatbot and robot).

(H3) Personality characteristics (particularly conscientiousness and open-mindedness), social desirability of responding, and technology affinity will predict satisfaction with the interview and communication

(H4) Willingness to repeat the interview with the same type of conversation they had just completed, in their assigned condition (psychiatrist-, chatbot-, or robot-delivered), will not significantly differ sbetween the three conditions, thus making the implementation of technology-mediated solutions feasible from the perspective of children and adolescents

(H5) Expert evaluations of the screening interview's clinical value will show generally favorable ratings across all conditions (above scale midpoint),with slightly favorable rating towards human-led approaches across most clinical utility dimensions.

The findings of this study could inform the development of accessible, self-administered screening tools that maintain clinical relevance while increasing access and reducing barriers to mental health assessment for youth populations, particularly in underserved settings where specialist access is limited.

## 2. Methods

### 2.1 Study design

In this research we applied a sequential approach consisting of three interconnected studies designed to evaluate a technology-mediated screening interview for identifying markers of mental health disorders in children and adolescents. The aim of *Study 1* was to establish content validity of the screening interview questions through expert evaluation of relevance and clarity, ensuring age-appropriateness for the 10–19 year target population. The aim of *Study 2* was to evaluate feasibility, user satisfaction, and willingness to engage with the screening interview across different modalities, and to examine factors impacting these outcomes. The aim of *Study 3* was to assess the clinical utility and value of the screening interview from expert perspectives, comparing effectiveness across different administration modalities. The study was registered under the ISRCTN Registry (ISRCTN68006163, registered 16/04/2025). The authors report no deviations to the registered protocol.

**2.1.1 Study 1: Content validity of the screening interview.** Study 1 was designed as a cross-sectional survey. A brief screening interview was co-developed with pediatric psychiatrists and psychologists to elicit contextual and observable information that can complement standardized screening tools in routine clinical practice (e.g., clarifying situational context, onset/timeline, and behavioral/interactional cues). The initial interview consisted of eight open-ended questions developed in collaboration with pediatric psychiatrists and psychologists to reflect a standard introductory clinical interview. The questions are presented in Table 1. Based on expert feedback, some questions were subsequently refined, resulting in the final 10-question version used in Study 2. These are presented Table 2. Participants were recruited among the employees of the University Medical Centre Maribor and researchers' contacts. Potential participants were sent an e-mail invitation including an explanation of the study and a link to the online questionnaire in Slovenian language (set up at https://1ka.arnes.si/). The first page of the online questionnaire explained the study and nature of participant's involvement along with their rights and the inclusion criteria for participation (i.e., expertise in the field of psychiatry or child and adolescent psychiatry or clinical psychology and treating individuals aged 10–19). By clicking on the "next page" button, participants confirmed their informed consent and proceeded to the second page which provided instructions for completing the questionnaire and included a video example of an interview to help better illustrate the nature of the interview questions they were about to evaluate. Study 1 applied a structured interview. The full set of questions asked to the participants is provided in Supplementary Information, S1 Table. Participants evaluated the interview questions and provided their demographic information. Participation in the study took on average approximately 9 minutes.

**2.1.2 Study 2: Feasibility and acceptability from youths' perspective.** Study 2 was designed as independent measures experimental study with three conditions. Participants were recruited via the Unit for Pediatric and Adolescent Psychiatry and randomly allocated to study using a computer-generated randomization sequence. i.e., prior to study initiation, sequential participant identifiers (pseudo-IDs) were randomly assigned to one of three conditions by MS Excel's RAND function by a researcher not involved in the execution of the experiments. Upon obtaining informed consent and/or verbal assent to participate in the study, participants received the next available pseudo-ID, ensuring allocation concealment until the point of assignment. This procedure resulted in three balanced groups:

- Psychiatrist condition (n = 35, 33.0%): Interview conducted by child and adolescent psychiatrist.

**Table 1. Content validity of interview questions.**

| Interview questions | Purpose of the questions | CVR | CVI[a] |
|---|---|---|---|
| 1. Can you tell me about yourself, so I can get an idea of who you are? | *To gain insights into self-perception, personality traits, and emotional expression.* | 0.71 | 0.94 |
| 2. How would you describe yourself to someone who is just getting to know you? Include your feelings, interests, and relationships in the description as well. | *To gain insights into self-perception, personality traits, and emotional expression.* | 0.80 | 0.89 |
| 3. Can you share what goes through your mind on a regular day, including any persistent negative thoughts or worries? | *To explore cognitive patterns and identify potential symptoms of anxiety or depression.* | 0.81 | 0.94 |
| 4. How would you describe your mood and energy levels in recent weeks? Have you noticed any changes affecting your daily life, work or school, family, or friends? | *To assess mood, energy, and the impact of emotional well-being on daily functioning.* | 0.81 | 0.94 |
| 5. Can you describe your usual response to difficult situations, especially in relation to others? | *To identify potential symptoms and behaviors associated with potential disorders.* | 0.71 | 0.94 |
| 6. When you're not feeling well, do you engage in any activities or hobbies? Does this help you feel better? | *To explore changes in interests and potential markers of mood disorders.* | 0.90 | 1.00 |
| 7. Is there anything specific that you find really fascinating or intriguing lately? And how did that make you feel? | *To explore changes in interests and potential markers of mood disorders.* | 0.81 | 1.00 |
| 8. How do you cope with stress or situations that overwhelm you? Can you recall any recent achievement or area you are proud of? | *To gauge coping strategies and assess self-esteem.* | 0.81 | 0.94 |

*Notes.* [a] CVI is calculated on the data from 18 participants, as three participants did not provide answers to evaluate the clarity of the interview questions.

PLOS Digital Health

**Table 2. Final interview questions.**

| Original question | Final questions |
|---|---|
| 1.* | 1. Can you tell me about yourself, so I can get an idea of who you are? |
| 2.* | 2. How would you describe yourself to someone who is just getting to know you? Can you describe your hobbies, relationship with your family and friends and emotions that you most often experience? |
| 3. | 3. Can you share what goes through your mind on a regular day, including any persistent negative thoughts or worries? |
| 4.* | 4. How would you describe your mood and energy levels in recent weeks? |
| 4.* | 5. Have you noticed any changes affecting your daily life, work or school, family, or friends? |
| 5.* | 6. How do you usually react in unpleasant and challenging situations, especially those that happen in relationships with others? |
| 6.* | 7. What do you do when you don't feel okay? Do you engage in some activity or hobby? How does this impact your mood? |
| 7. | 8. Is there anything specific that you find really fascinating or intriguing lately? And how did that make you feel? |
| 8.* | 9. Do you ever feel overwhelmed? What do you do then? |
| 8.* | 10. Can you think of any recent accomplishments that you are proud of? Are you really good at anything? |

*Notes.* * Original question was modified based on the experts' comments.

- Chatbot condition (n = 34, 32.1%): Self-administered interview using tablet-based chatbot application.

- Robot condition (n = 37, 34.9%): Interview facilitated by socially assistive humanoid robot (Pepper model).

A similar study procedure was followed by all participants. First, they were welcomed by a researcher and again briefly informed about the study's purpose and the procedure of the experiment. Second, they completed baseline questionnaires (personality assessment, technology affinity) and participated in video-recorded screening interviews according to their assigned condition. In Chatbot condition, a participant was alone in the room together with a tablet with a chatbot app. Participants read the chatbot's questions via the tablet and responded verbally. Their spoken answers were recorded but did not require any typing on the device. In the Robot condition, a participant was alone in the room together with the robot. Robot, being remotely controlled by the operator, guided the interview and posed questions that the participant responded to. During the robot-facilitated interview, participants listened to the to the robot's questions and responded verbally to all questions. No typing or interaction with additional devices was required. In both Chatbot and Robot conditions, a researcher was present in another room and monitored participants to ensure their safety and provide help in case of emergency. At the end all participants completed post-interview evaluations (satisfaction with the communication in the screening interview, willingness to repeat, social desirability measures).

Finally, participants were fully debriefed on the study's purpose and procedure, as they were deceived regarding the presence of the researcher in another room monitoring the interview (and operating the robot in the Robot condition) in the Chatbot and Robot condition, and on the existence of two other experimental conditions. They were also offered support if needed.

**Blinding Procedures:** Complete blinding of participants was not feasible due to the distinct nature of each intervention modality. Namely, the patients could distinguish clearly between receiving robot-, tablet- and human-delivered conversation. However, participants were not informed about the specific content, extent, or comparative aspects of interventions

in other study arms prior to their participation. They were informed only about the modality they would experience. This approach was introduced to minimize the cross-modality comparison effects generated through potential demand characteristics, disappointment or curiosity about 'missing' conditions.

**Experimental conditions:**

*Psychiatrist Condition:* Participants engaged in face-to-face interviews conducted by board-certified pediatric psychiatrists in clinical office settings. The interview followed a conversational format, with psychiatrists adapting their pace and language to match participants' age and comprehension levels while maintaining adherence to the structured 10-question protocol.

*Chatbot Condition:* Participants completed self-administered interviews using a tablet-based chatbot application (see Fig 1). The interface presented questions sequentially in a user-friendly format, allowing participants to respond at their own pace in a private room setting with human supervision hidden from participants. A RASA-based chatbot (v3.6.21) [50] orchestrated structured interactions between participants employing a deterministic, rule-based approach where the interview was designed as a white-box model with well-defined storyline and finite inputs/outputs. The interaction session was structured as a predetermined storyline aligned with the adhering to the structured 10-question protocol, prioritizing transparency, safety, and clinical accuracy over conversational flexibility.

*Robot Condition:* The intervention utilized the Pepper robot (SoftBank Robotics, hardware version: 1.8, software version: NAOqi 2.5 SDK), a 120 cm tall humanoid robot known for its affordability and popularity in human-robot interaction studies (see Fig 2). We have reutilized a platform developed for robot-assisted nursing [51]. The platform recognizes Pepper's

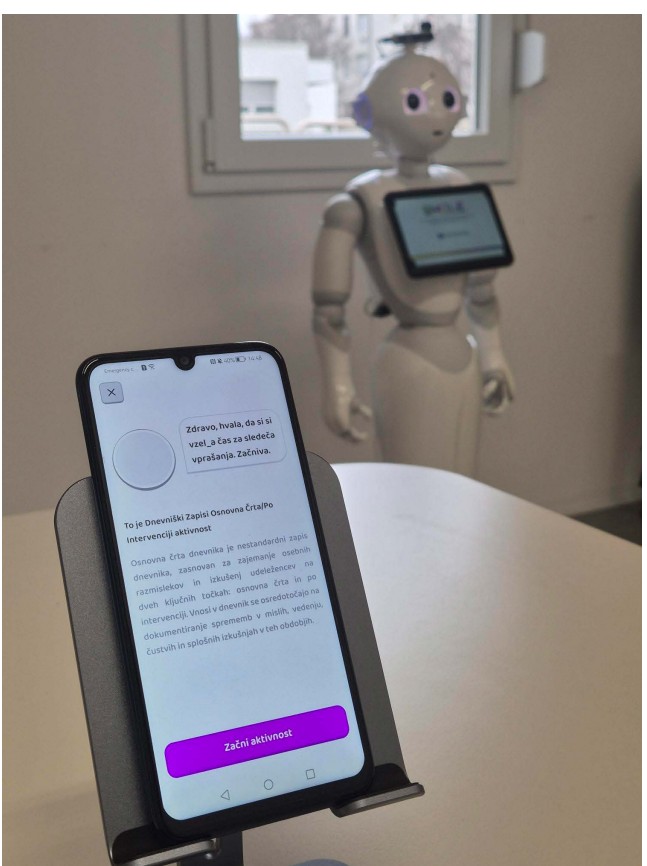

**Fig 1. Tablet-mediated intervention.**

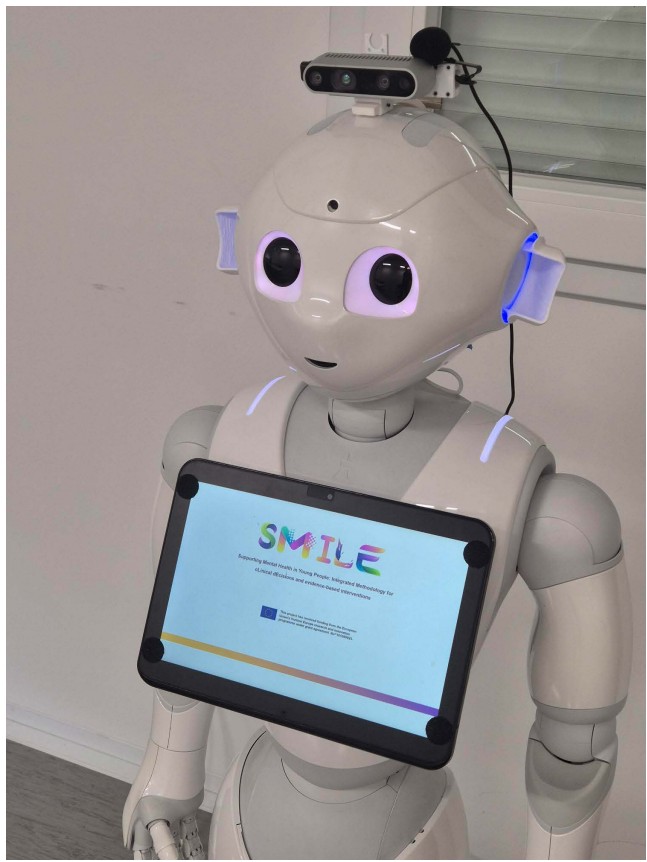

**Fig 2. Robot-mediated intervention.**

limitations in advanced communicative capabilities and support for non-mainstream languages and implements customizations to optimize performance for this study. The socially assistive robot (SAR) intervention employed symmetric interaction architecture [52]. In addition to RASA-based chatbot (v3.6.21) driving the conversation, the PLATTOS speech synthesis system [53] was implemented to overcome Pepper's limitations in speech production, particularly in Slovenian language. This end-to-end multilingual and multimodal text-to-conversation synthesis system utilized a multi-stage architecture including speech production and gesture synthesis. The ASR SPREAD (Automatic Speech Recognition) system [52] enabled the robot to understand and respond to participants' verbal inputs.

**2.1.3 Study 3: Clinical utility of the screening interview: expert evaluation.** Study 3 was designed as expert evaluation study using video analysis. Participants were recruited via the Unit for Pediatric and Adolescent Psychiatry, University Medical Centre Maribor. Experts independently evaluated screening interview videos across all three conditions using a structured questionnaire. While the experts were not directly revealed the modality used for delivery, full blinding of expert evaluators was not feasible as the video recordings clearly revealed the intervention type (psychiatrist presence, tablet interface, or robot interaction).

## 2.2 Measures

**2.2.1 Study 1: Assessment of content validity of the screening interview.** *Screening interview questions.* Interview questions to be evaluated in this study were developed in collaboration with experts, i.e., pediatric psychiatrists

and psychologists (authors of this paper). They were formed to loosely follow the standard initial conversation that experts hold with patients in psychiatric and child and adolescent psychiatric departments, with the aim to be used in further research on capuring the digital markers of various mental health disorders.

***Content validity.*** Content validity of the interview questions was assessed via two scales used to evaluate the relevance and clarity of each of the eight interview questions. On the relevance scale, participants rated each interview question by assigning them one of the following values [54]: 1 (*"The item is essential"*), 2 (*"The item is useful but not necessarily essential"*), or 3 (*"The item is not needed"*). On the clarity scale, participants rated each interview question by assigning them one of the following values: 1 (*"The item is clear"*), 2 (*"The item should be partially modified to achieve the desired clarity"*), 3 (*"The item should be substantially modified to achieve the desired clarity"*), or 4 (*"The item is completely unclear"*) [55]. Additionally, a space for comments and suggestions for improvement for each of the interview questions was provided.

***Demographic questions.*** Participants also provided information on their demographic characteristics, i.e., occupation, whether they encounter individuals aged 10–19 in their work, years of professional experience, type of organization they are employed in, their highest level of completed education, and gender.

### 2.2.2 Study 2: Feasibility and User Experience Assessment. *Screening interview*. Participants answered the ten final questions of the screening interview from Study 1 (Table 2).

***Social desirability of responding.*** For the assessment of social desirability of responding, a short version of the Social Desirability Response Questionnaire (BIDR-16) [56] was used. BIDR-16 consists of two subscales: Self-deceptive enhancement (eight items; e.g., *"I never regret my decisions."*; one item was omitted in the present study due to its inappropriate content for the target population, i.e., *"I have sometimes doubted my ability as a lover."*), and Impression management (eight items; e.g., *"I never cover up my mistakes."*). Items are rated on a seven-point Likert scale (1 – *Not true*, 4 – *Somewhat true*, 7 – *Very true*). Validation studies have shown acceptable internal consistency of the subscales (Self-deceptive enhancement: $\alpha = .63$ -.82; Impression management: $\alpha = .66$ -.73), with similar values found in the present study (Self-deceptive enhancement: $\alpha = .66$; Impression management: $\alpha = .73$).

***Personality.*** Personality dimensions were assessed with the Big Five Personality Inventory – 2, Extra Short Form (BFI-2-XS) [57]. Each of the five dimensions, i.e., Extraversion (e.g., *"Is dominant, acts as a leader."*), Agreeableness (e.g., *"Is compassionate, has a soft heart."*), Conscientiousness (e.g., *"Is reliable, can always be counted on."*), Negative emotionality (e.g., *"Worries a lot."*), and Open-mindedness (e.g., *"Is fascinated by art, music, or literature."*), is assessed with three items that are rated on a five-point Likert scale (1 – *Disagree strongly*, 5 – *Agree strongly*). The validation study, presented in the Soto & John (2017) [57] has shown acceptable internal consistency of the subscales (Extraversion: $\alpha = .63$ -.66; Agreeableness: $\alpha = .49$ -.55; Conscientiousness: $\alpha = .55$ -.61; Negative emotionality: $\alpha = .69$ -.73; Open-mindedness: $\alpha = .57$ -.58). Similar values were found in this study (Extraversion: $\alpha = .68$; Agreeableness: $\alpha = .44$; Conscientiousness: $\alpha = .60$; Negative emotionality: $\alpha = .63$; Open-mindedness: $\alpha = .62$).

***Technology Affinity.*** Technology affinity was assessed with the Inclusion of Technology Affinity in Self scsale (ITAS) [58]. It consists of a single item (i.e., *"Please consider the following circles. What affinity do you have for technical systems?"*) that is answered on a seven-point scale where each point on a scale represents a graphical overlap between two circles representing a person responding and technical systems, from circles not overlapping at all to circles completely overlapping. The minimum estimated reliability (as correlation of ITAS and another reference questionnaire) of the ITAS in the validation study was acceptable ($r_{yy} = 0.64$).

***Satisfaction with the screening interview.*** General satisfaction with the screening interview was assessed using a single item (i.e., *"How satisfied are you with the conversation you participated in?"*) answered on a five-point scale (1 – *Very dissatisfied*, 3 – *Neither satisfied nor dissatisfied*, 5 – *Very satisfied*), developed by the authors, following the findings of previous studies [59–61].

*Satisfaction with the communication in the screening interview.* Satisfaction with the communication in the screening interview was assessed with a Communication subscale of a Generic Medical Interview Satisfaction scale (G-MISS) [62] that measures participants' experience with the communication aspect of the health interview. The subscale consists of six items (e.g., *"The doctor seemed warm and friendly to me."*) that are measured on a five-point scale (1 – *Strongly disagree*, 5 – *Strongly agree*). For the use in the present study, the items were adapted to appropriately refer to the non-human moderators in Robot (e.g., *"The robot seemed warm and friendly to me."*) and Chatbot (e.g., *"The chatbot seemed warm and friendly to me."*) conditions. The validation study of the original G-MISS questionnaire showed acceptable reliability of the Communication subscale ($\alpha = .71 -.75$). In the present study, values of internal consistency were similar in the Psychiatrist and Chatbot conditions ($\alpha = .59$ and $\alpha = .70$, respectively), however they were lower in the Robot condition ($\alpha = .38$).

**Willingness to conduct the screening interview.** Two items were developed by the authors to assess the willingness to conduct the screening interview (i.e., *"If you were to experience periods when you felt unwell or noticed something about yourself that concerned you, would you be willing to have such a conversation again?"*), and the frequency of doing so (i.e., "*If you were to experience periods when you felt unwell or noticed something about yourself that concerned you, would you be willing to have such a conversation again?*"). The first item was answered on a three-point scale (1 – *Yes*, 2 – *Maybe*, 3 – *No*), and the second one on a six-point scale (1 – *Once a week*, 2 – *Once a month*, 3 – *A few times a year*, 4 – *Once a year*, 5 – *Less than once a year*, 6 – *Not at all*), respectively.

**Demographic information.** Participants were asked to provide age, gender, and current education/schooling enrollment. Psychiatrists of the Unit for Pediatric and Adolescent Psychiatry also provided participants' existing psychiatric diagnoses.

Questionnaires not previously available in Slovenian (i.e., BIDR-16, and G-MISS Communication subscale) were translated using a translation and back-translation process. Two researchers independently translated the items into Slovenian, then a third combined their versions. A fourth researcher, unfamiliar with the originals, translated the merged version back into the source language. Any differences between the original and back-translated versions were discussed and resolved. All translators were highly proficient in both languages and had backgrounds in psychology or psychiatry and social science methodology.

**2.2.3 Study 3: Clinical utility assessment.** To assess experts' evaluations of the value of the screening interviews in improving clinical decision-making, we formulated eight questions (available in Supplementary Information, S8 Table) on three topics: (1) Relevance and completeness of information (1 item; coverage of psychosocial factors); (2) Clinical usefulness and significance for treatment planning (5 items; contribution to diagnosis accuracy, support for setting personalized treatment goals, aid in choosing appropriate interventions, suitability for ongoing progress monitoring, and usefulness in identifying high-risk individuals), and (3) Integration with clinical assessment (2 items; complement and enhancement of overall clinical evaluation, and alignment with other assessment methods). Questions were answered on a seven-point scale (1 – *Not at all*, 4 – *Unable to decide*, 7 – *Completely*).

### 2.3 Recruitment

**2.3.1 Recruitment and eligibility criteria for Study 1.** Participants were recruited among employees of University Medical Centre Maribor and researchers' contacts, between July 2024 and September 2024. The inclusion criteria were: (1) expertise and practice in the field of psychiatry, child and adolescent psychiatry, or clinical psychology, and (2) ability to evaluate interview questions for relevance and clarity. No specific exclusion were put in place.

**2.3.2 Recruitment and eligibility criteria for Study 2.** Participants were recruited via the University Division of Pediatrics, Unit for Pediatric and Adolescent Psychiatry of University Medical Centre Maribor, between October 2024 and March 2025. The inclusion criteria were the following: 1) being treated at the Unit for Pediatric and Adolescent Psychiatry (inpatient ward or outpatient clinic); 2) age between 10 and 19 years at the time of obtaining consent; 3) sufficient cognitive abilities to understand and respond to the screening interview questions; 4) fluency in Slovenian language to

properly understand and respond to screening interview questions; 5) signed informed consent for the participants of 15 years of age and older, and signed informed consent from parents or legal guardians for those younger than 15 and the participant's verbal assent; and 6) availability for the time required to complete all parts of the study. Participants were excluded if: 1) they had mild to severe intellectual development disorder that could impair their ability to understand or participate in the screening interview process; 2) were experiencing acute psychiatric conditions, e.g., life-threatening conditions or other states requiring immediate psychiatric intervention; 3) had sensory impairments that would prevent interaction with the chatbot, robot, or psychiatrist conducting the screening interview unless reasonable accommodations allowed for effective participations; or 4) had serious somatic conditions that could interfere with their ability to participate in the study or respond reliably.

**2.3.3 Recruitment and eligibility criteria for Study 3.** Participants were recruited within the University Division of Pediatrics, Unit for Pediatric and Adolescent Psychiatry of University Medical Centre Maribor, between April 2025 and May 2025. The inclusion criteria were: (1) Child and adolescent psychiatry specialists or residents with active practice, (2) Clinical experience in evaluating mental health interviews and (3) Ability to evaluate video recordings using structured assessment tools. No specific exclusion criteria were defined.

## 2.4 Statistical analysis

**2.4.1 Study 1: Content validity assessment of the screening interview items.** To evaluate the relevance of the interview questions, we calculated the content validity ratio (CVR; Lawshe, 1975). CVR can range from -1.00 to 1.00, and higher values indicate higher relevance of the questions. When more than half of respondents evaluate a question as "essential", the CVR is between 0.00 and 1.00. In this study, a cutoff of.68 for 21 respondents was used as a criterion [63]. To evaluate the clarity of the questions, we calculated the content validity index (CVI) [55];. When more than half of the respondents evaluate a question as only needing a partial modification, CVI is between.50 and 1.00, and in this study, a conventional cutoff of.80 was used as a criterion. Furthermore, we reviewed participants' comments and used them for interview questions' modifications. s

**2.4.2 Study 2: Feasibility and user experience assessment.** We first examined the database. As there were no missing values, no data needed to be imputed. Next, we calculated questionnaires' scores, descriptive statistics, tested the assumptions of planned analyses, and proceeded with analyses in IBM SPSS Statistics 29. First, we examined the internal consistency of measures with α coefficient. Then we proceeded with one-way analysis of variance (ANOVA) to examine the differences among conditions regarding the satisfaction with the screening interview and the communication, and willingness to repeat the interview. Lastly, we conducted two sets of multiple regression analyses. In the first set (Table 4), we examined the predictors of satisfaction with and willingness to repeat the interview outcomes while controlling for the condition participants were assigned to via hierarchical regression analyses in two steps. In the first step, we added dummy variables of experimental conditions to the model (i.e., Psychiatrist vs Chatbot, and Psychiatrist vs Robot), and in the second step, we added basic demographic characteristics (i.e., gender, age) and other measured variables (i.e., personality dimensions, social desirability of responding, and technology affinity).

**2.4.3 Study 3: Clinical Utility Assessment.** First, we examined the database. Due to the low quality of some video recordings, not all of them were suitable to be evaluated by the experts and were therefore excluded from this study (of total 106 screening interview videos, 13 (12.26%) were excluded). In the final database, 1.34% of data were missing. As they were not missing completely at random (Little's MCAR test, $\chi^2(14) = 28.521$, $p = .012$), they were not replaced. Second, we examined the potential differences in mean evaluations provided by four experts (reported in Supplementary Information, S9 Table). As they did not differ significantly, we proceeded to calculate descriptive statistics and test the assumptions of planned analyses. Finally, we conducted ANOVAs in IBM SPSS Statistics 29 to examine the differences among interview videos conducted in three conditions in experts' evaluations.

## 2.5 Consent

For the study a separate ethics approval was received from the Medical Ethics Commission of the University Medical Center Maribor, Ljubljanska ulica 5, 2000 Maribor, Slovenia; (+386 (0)2 321 2489; eticna.komisija@ukc-mb.si), on 4th July 2024, registration number: UKC-MB-KME-32/24. Informed consent was obtained from all the subjects (and where relevant their legal guardians) involved in the study in paper format. This study is registered at ISRCTN Registry, Clinical trial number: ISRCTN68006163 (https://doi.org/10.1186/ISRCTN68006163), registered on 16/04/2025.

## 3. Results

### 3.1 Study 1: Content validity assessment

**3.1.1 Participant characteristics.** A total of 76 participants clicked on the survey, and 40 (53.6%) began filling it in. 18 (45.0%) participants were excluded due to non-completion of the survey, and an additional one (2.5%) due to non-compliance with the inclusion criteria. The final sample consisted of 21 participants. 10 (47.6%) of them were child and adolescent psychiatry residents, followed by 5 (23.8%) child and adolescent psychiatry specialists, 2 (9.5%) clinical psychology residents, 2 (9.5%) psychologists, one (4.8%) psychiatry specialist, and one (4.8%) psychiatry resident. 9 (42.9%) participants were employed at the primary level of healthcare system, 8 (38.1%) at the tertiary, one (4.8%) at the secondary level, while 2 (9.5%) participants were cycling at all three levels of the healthcare system due to their residency requirements. Participants had from 1 to 32 years of work experience (M = 5.4, SD = 6.5). All the participants except one (4.8%) that did not respond held a second-cycle (Master's-level) degree according to the Bologna system or an equivalent degree (n = 29, 95.2%). 16 (76.2%) participants were female, 4 (19.1%) male, while one (4.8%) did not respond to the question regarding gender.

**3.1.2 Content validity and final interview questions.** Results showed that all interview questions passed Content validity index (CVI) and Content validity ratio (CVR) criteria (Table 1), i.e., none of them needed to be modified based on the experts' quantitative evaluations of questions relevance and clarity.

Experts also provided some comments on the questions and suggestions for their improvement, in general expressing the need for clearer, more specific, and simpler and concrete questions and the importance of avoiding vagueness or compound wording (full summary of the comments is available in S1 Table. Therefore, some of the questions were slightly modified or split into two questions, resulting in overall ten final interview questions (Table 2).

### 3.2 Study 2: Feasibility of AI- and Human-led screening interview – youth perspective

Descriptive statistics of the central variables and correlations between them for the total sample and by experimental conditions are presented in S3 Table, S4 Table, S5 Table and S6 Table. In the following sections, we describe the participant characteristics and the results of comparisons of satisfaction with and willingness to repeat the pre-screening interview among the three experimental conditions, and factors associated with these outcomes.

**3.2.1 Participant characteristics.** The final sample consisted of 106 participants aged between 10 and 19 years (M = 15.8, SD = 1.78). The majority of participants identified as female (n = 72, 67.9%), with 29 identifying as male (27.4%), two as other (1.9%), and three participants (2.8%) choosing not to disclose their gender. Regarding educational status, 29 participants (27.4%) were enrolled in primary school, 74 (69.8%) attended secondary or high school, and three participants (2.8%) were not enrolled in any form of education at the time of the study. Across participants, the number of diagnoses per person ranged from 0 to 5 (M = 1.79, SD = 0.97), with 10 participants (9.4%) had no prior diagnosis recorded. The most prevalent diagnoses observed in the study sample were within the following ICD-10 categories: F41 – Other anxiety disorders (n = 20, 18.9%), F50 – Eating disorders (n = 18, 17.0%), F60 – Specific personality disorders (n = 17, 16.0%), F92 – Mixed disorders of conduct and emotions (n = 17, 16.0%), F32 – Depressive episode (n = 15; 14.2%), F90 – Attention-deficit hyperactivity disorders (n = 14, 13.2%), and R45 – Symptoms and signs involving emotional

state (n = 14, 13.2%). Other diagnoses appeared less than 10 times in the total sample. Detailed information on diagnoses is provided in S2 Table. Participants were randomly assigned to one of three conditions, i.e., Psychiatrist (n = 35, 33.0%), Chatbot (n = 34, 32.1%), and Robot (n = 37, 34.9%).

While the initial sample size calculation indicated 90 participants would be sufficient to detect a medium effect size (d = 0.5) with α = 0.05 and power of 0.80, we recruited 106 participants to account for potential dropouts, ensure balanced group allocation across the three experimental conditions, and maintain adequate statistical power despite the exclusion of some video recordings due to technical quality issues in the expert evaluation phase.s

**3.2.2 Satisfaction with and willingness to repeat the pre-screening interview among three conditions.** We conducted ANOVA analysis for satisfaction and willingness outcomes; detailed results are presented in Table 3 and described below.

The significant ANOVA results were followed up with post-hoc tests using Tukey's HSD test. For satisfaction with the interview, post-hoc comparisons indicated that the mean scores of Psychiatrist and Chatbot conditions differed significantly (p < .05, 95% CI = [.14, 1.44]), i.e., satisfaction with the interview was higher in the Psychiatrist than in the Chatbot condition. However, there were no significant differences in satisfaction with the interview between the mean scores of Psychiatrist and Robot conditions (p = .384, 95% CI = [-.28,.99]), and between the mean scores of Chatbot and Robot conditions (p = .239, 95% CI = [-1.08,.20]).

For satisfaction with communication in the interview, post-hoc comparisons indicated that the mean scores of Psychiatrist and Chatbot conditions differed significantly (p < .001, 95% CI = [.54, 1.29]), and so did the mean scores of Psychiatrist and Robot conditions (p < .001, 95% CI = [0.35, 1.10]). In both cases, the participants in the Psychiatrist condition were more satisfied with the communication than they were in the Chatbot and Robot conditions, respectively. However, there were no significant differences in satisfaction with communication between the mean scores of Chatbot and Robot conditions (p = .457, 95% CI = [-.56,.19]).

**3.2.3 Factors associated with the satisfaction and willingness to conduct the screening interview.** The model for satisfaction with communication explained a significant portion of the variance (i.e., 32%), as outlined in Table 4. In step 1 (i.e., S1) of the multiple regression analyses there were only the experimental conditions included. The inclusion of additional variables in step 2 (i.e., S2) also explained a significant portion of the variance above the first model (i.e., additional 14%). Of the variables included in the second step, conscientiousness ($\beta$ = .28, p = < .01) and open-mindedness ($\beta$ = .24, p = < .01) significantly predicted satisfaction with communication in the interview.

A regression model for satisfaction with the interview also explained a significant portion of variance in the first step (i.e., 10%), while variables added in the second step did not explain a significant amount of variance above the first model (i.e., additional 12%). In the analyses for the willingness to repeat the interview outcomes, none of the models explained a significant amount of variance.

**Table 3. Differences in satisfaction with and willingness to repeat the pre-screening interview among the three conditions.**

| | Psychiatrist | | Chatbot | | Robot | | ANOVA | |
|---|---|---|---|---|---|---|---|---|
| | *M* | *(SD)* | *M* | *(SD)* | *M* | *(SD)* | *F* | *η2* |
| Satisfaction with the interview | 4.06 | (1.11) | 3.26 | (1.26) | 3.70 | (1.02) | 4.230* | 0.076 |
| Satisfaction with communication | 3.92 | (0.64) | 3.01 | (0.75) | 3.20 | (0.60) | 18.528*** | 0.265 |
| Willingness to repeat the interview | 1.49 | (0.61) | 1.71 | (0.68) | 1.73 | (0.77) | 1.341 | 0.025 |
| Willingness to repeat the interview – frequency | 2.86 | (1.17) | 3.09 | (1.51) | 3.11 | (1.65) | 0.324 | 0.006 |

*Notes*. For all comparisons: $F_{(2, 103)}$. * p < .05, ** p < .01, *** p < .001. [a] Lower score indicates higher willingness of conducting the screening interview. [b] Lower score indicates willingness to conduct the screening interview with higher frequency.

**Table 4. Predictors of satisfaction with and willingness to repeat the screening interview.**

| | Satisfaction with the interview | | Satisfaction with communication | | Willingness to repeat the interview | | Willingness to repeat the interview - frequency | |
|---|---|---|---|---|---|---|---|---|
| | S1 | S2 | S1 | S2 | S1 | S2 | S1 | S2 |
| Psychiatrist vs Chatbot | -0.37*** | -0.34*** | -0.62*** | -0.61*** | 0.17 | 0.19 | 0.07 | 0.11 |
| Psychiatrist vs Robot | -0.14 | -0.12 | -0.50*** | -0.45*** | 0.14 | 0.18 | 0.07 | 0.12 |
| Gender | | 0.01 | | 0.10 | | 0.02 | | 0.09 |
| Age | | -0.14 | | -0.06 | | 0.10 | | 0.09 |
| Extraversion | | -0.07 | | 0.00 | | -0.06 | | 0.10 |
| Agreeableness | | 0.16 | | -0.06 | | -0.03 | | 0.09 |
| Conscientiousness | | 0.16 | | 0.28** | | -0.12 | | -0.10 |
| Negative emotionality | | -0.10 | | -0.04 | | -0.08 | | -0.01 |
| Open-mindedness | | 0.20* | | 0.24** | | -0.10 | | -0.25 |
| Self-deceptive enhancement | | -0.08 | | -0.17 | | -0.05 | | -0.02 |
| Impression management | | -0.06 | | 0.05 | | 0.12 | | 0.02 |
| Technology affinity | | 0.06 | | 0.00 | | -0.07 | | -0.04 |
| $R^2$ | 0.10 | .22 | .32 | .46 | .02 | .08 | .01 | .09 |
| $F^a$ | 5.52** | 2.05* | 22.70*** | 6.15*** | 1.22 | .65 | .23 | .74 |
| $\Delta R$ | .10 | .12 | .32 | .14 | .02 | .06 | .01 | .09 |
| $\Delta F^b$ | 5.52** | 1.32 | 22.70*** | 2.25* | 1.22 | .55 | .23 | .84 |

*Notes.* Standardized betas are reported. N = 101 as we only included those participants that identified as male (0) or female (1) in these analyses. S1 = Step 1, S2 = Step 2. [a] Degrees of freedom were 2, 98 in Step 1; 12, 88 in Step 2. [b]Degrees of freedom were 2, 98 in Step 1; 10, 88 in Step 2. * $p < .05$, ** $p < .01$, *** p < .001

In the second set of regression analyses (reported in S7 Table), we only included the samples of Chatbot and Robot conditions, as we were interested in whether specific participants' characteristics predict satisfaction with and willingness to repeat the interview that was conducted with the help of technology. Results showed that no models explained a significant amount of variance.

### 3.3 Study 3: Feasibility of AI- and Human-led screening interview – experts' perspective

**3.3.1 Participant characteristics.** A total of 4 experts were involved in evaluating the screening interview videos, recorded in Study 2. Two of them (50.0%) were child and adolescent psychiatry residents, and two (50.0%) child and adolescent psychiatry specialists. They had 5–25 years of work experience (M = 12.3, SD = 9.2). The experts used a battery of questions outlined in S8 Table. One expert evaluated 8 screening interviews (by conditions: Chatbot = 4, Robot, 4), the second one evaluated 17 interviews (by conditions: Psychiatrist: 6, Chatbot: 6, Robot: 5), the third evaluated 17 interviews (by conditions: Psychiatrist: 4, Chatbot: 8, Robot: 5), and the fourth expert evaluated 51 screening interviews (by conditions: Psychiatrist: 17, Chatbot: 13, Robot: 21).

**3.3.2 Analyzing experts' evaluations of the screening-interview videos among three conditions.** First, we calculated mean values of experts' evaluations of the interview video-recordings according to three conditions and variables assessed. These revealed that almost all evaluations (i.e., of all variables in almost all interview conditions) were above the scale's mid-point, suggesting that, in general, the value of such interview was evaluated favorably. Next, we conducted a one-way ANOVA comparissng experts' evaluations of the screening-interview videos across three interviewer modalities: Psychiatrist, Chatbot, and Robot.s

The results showed a significant effect of these conditions for only one variable, i.e., alignment with other assessment methods. For all other evaluation variables, there were no significant differences among the conditions (see Table 5 and S9 Table).

The significant ANOVA result was followed up with post-hoc comparison using Tukey's HSD test. For alignment with other assessment methods, only videos of Psychiatrist and Robot conditions were evaluated significantly differently ($p < .05$, 95% CI = [.16, 1.78]), i.e., videos of Psychiatrist condition were evaluated more favorably than videos of Robot condition. However, there were no significant differences in evaluations regarding alignment with other assessment methods between the mean scores of Psychiatrist and Chatbot conditions videos ($p = .552$, 95% CI = [-.48, 1.22]), and between the mean scores of Robot and Chatbot conditions videos ($p = .168$, 95% CI = [-.19, 1.38]).

## 4. Discussion

The studies presented in this paper demonstrate that technology-mediated screening interviews represent a feasible and clinically valuable approach for mental health assessment in children and adolescents. Digital health solutions, including technology-mediated screening, represent a critical pathway for expanding treatment access and increasing the range of available interventions for adolescent mental health [18], particularly in settings where traditional face-to-face assessment remains inaccessible due to workforce shortages. Moreover, while gold standard, self-report measures, such as, PHQ-9, GAD-7, MINI-KID and K-SADS, can be influenced by recall and response biases, the conversational screening may help provide additional context and clarify responses and guide follow-up questioning [64,65]. While the study did not evaluate impacts on diagnostic accuracy or clinical outcomes, the demonstrated, content validity, feasibility and clinical utility of technology-mediated screening interviews show a clear potential of the screening interview to address a fundamental challenge in youth mental health: the critical shortage of child and adolescent mental health specialists that results in significant barriers to accessing timely, specialized care [4–6].

The results support our initial hypothesis (H1), regarding content validation. The analysis confirms that questions of the screening interview surpassed the minimum criteria for the indices of relevance and clarity, pointing to acceptable content validity according to established psychometric criteria. The validation ensures the clinical relevance and age-appropriateness of the defined protocol for capturing essential mental health markers in children and adolescents. The validation is particularly relevant since half of all lifetime mental health conditions emerge by age 14 [2,3], making effective screening tools for this population essential for early intervention efforts.

The feasibility testing in Study 2 supported the second hypothesis (H2). As initially predicted, psychiatrist-led interviews achieved higher satisfaction ratings compared to chatbot conditions for general interview satisfaction and higher

**Table 5.  Differences in evaluations of interviews in three conditions by experts.**

|  | Psychiatrist | | Chatbot | | Robot | | ANOVA | |
|---|---|---|---|---|---|---|---|---|
|  | M | (SD) | M | (SD) | M | (SD) | F | η2 |
| Coverage of psychosocial factors[a] | 4.30 | (1.49) | 4.80 | (1.42) | 4.03 | (1.67) | 2.055 | 0.044 |
| Contribution to diagnosis accuracy[b] | 4.52 | (1.31) | 4.42 | (1.65) | 4.29 | (1.47) | 0.191 | 0.004 |
| Support for setting personalized treatment goals[b] | 4.56 | (1.16) | 4.55 | (1.63) | 4.14 | (1.85) | 0.712 | 0.016 |
| Aid in choosing appropriate interventions[b] | 4.59 | (1.12) | 4.45 | (1.48) | 4.14 | (1.75) | 0.744 | 0.016 |
| Suitability for ongoing progress monitoring[b] | 5.11 | (1.19) | 5.19 | (1.62) | 4.54 | (1.63) | 1.808 | 0.039 |
| Usefulness in identifying high-risk individuals[b] | 5.26 | (1.10) | 5.16 | (1.42) | 4.74 | (1.79) | 1.088 | 0.024 |
| Complement and enhancement of overall clinical evaluation[b] | 4.41 | (1.34) | 4.48 | (1.50) | 3.89 | (1.62) | 1.544 | 0.033 |
| Alignment with other assessment methods[c] | 5.33 | (1.05) | 4.96 | (1.06) | 4.36 | (1.54) | 4.281* | 0.096 |

Notes. * $p < .05$. [a] F (2, 89), [b] F (2, 90), [c] F (2, 81). Numbers of evaluated videos (Psychiatrist condition, n = 27; Chatbot, n = 30; Robot, n = 35) are smaller than the N of participants in each condition in Study 2 due to low quality of recordings in some cases.

satisfaction with communication compared to both technological conditions (chatbot and robot). It is however worth mentioning that while robot conditions showed numerically higher satisfaction than chatbot conditions, these differences did not reach statistical significance. Since Study 2 used a between-subject design, the individual-difference variation between randomized groups could have increased error variance and reduced statistical power. These may also partially account for some observed differences in satisfaction. Thus, modality effects on satisfaction with interview, should be interpreted accordingly.

These findings align with limited research addressing human versus technology comparisons in child and adolescent populations [66–69], while contrasting with studies showing comparable or superior outcomes for digital mental health interventions [20–22,27,29]. The preference for human interaction likely reflects developmental needs for authentic human connection during mental health interventions [69], as children and adolescents have unique communication requirements that typically favor human interaction [66]. However, the critical finding supporting the feasibility of technology-mediated approaches is that willingness to repeat screening did not differ significantly between conditions, distinguishing satisfaction preferences from acceptance and practical utility. This suggests that while youth may prefer human interaction when available, they remain equally open to technology-assisted screening when human specialists are unavailable [6,11,67,70]. Furthermore, the role of individual personality factors in predicting satisfaction, specifically conscientiousness for satisfaction with communication, and open-mindedness for both, general satisfaction and satisfaction with communication, as observed in Study 2, indicates potential for personalized screening delivery approaches.

Our analysis also supports (H3). Namely, the results show that personality characteristics predicted satisfaction outcomes. Specifically, conscientiousness and open-mindedness emerged as significant predictors of satisfaction with communication across all conditions. The findings indicate the potential for personalized screening delivery approaches, where individuals higher in conscientiousness, characterized by self-discipline and goal-directed behavior, and open-mindedness, associated with willingness to engage with novel technologies, show greater satisfaction with structured interview processes [30–35]. The equivalence in willingness suggests that barriers to technology-mediated screening are not fundamentally related to user acceptance but may instead involve implementation factors, accessibility, or system design considerations. The finding supports the viability of technology-mediated screening as a complementary approach to traditional assessment methods [12,14]. While our study did not include a dedicated self-disclosure scale, communication satisfaction and willingness-to-repeat scores in the chatbot and robot conditions indicate that many adolescents felt comfortable speaking and would be willing to engage in such interviews again. This pattern offers preliminary, indirect support for the notion that technology-based pre-screening can support adolescents in sharing sensitive information, while underlining the need for future studies that incorporate dedicated, validated self-disclosure measures. The observed pattern is broadly in line with previous work indicating that technology-mediated or electronic assessments can provide a comfortable context for adolescents to share sensitive information [71].

(H4) was also fully supported by the results. Namely we did not find significant differences regarding willingness to repeat the interview between the three conditions. The distinction between satisfaction and willingness represents a key finding that also supports H2, i.e., while user preferences may favor human interaction, practical acceptance of technology-mediated alternatives remains high when needed. This finding aligns with research demonstrating that adolescents can adapt to different interview modalities despite having preferences [38], and supports evidence showing adolescent acceptance of digital mental health approaches [24,67]. Moreover, the results align with the literature analyzing successful factors for implementing electronic mental health screening in routine youth care [70]. It suggests that while delivery modalities may influence user experience, they do not fundamentally impair willingness to engage with screening processes. However, this must be considered alongside broader literature showing willingness does not automatically translate to equivalent implementation success [68]. Thus, while equivalence in willingness suggests that barriers to technology-mediated screening are not fundamentally related to user acceptance the implementation factors, accessibility, or system design considerations can still present implementation challenges in youth populations.

Finally, the expert clinical evaluation in Study 3 provided partial support for the fifth hypothesis (H5). Expert evaluations demonstrated generally favorable ratings across all clinical utility dimensions for all three modalities, with mean scores above the scale midpoint for every evaluated criterion. However, contrary to our prediction of slight preference for human-led approaches across most dimensions, significant differences emerged for only one criterion: alignment with other assessment methods, where psychiatrist-led interviews were rated more favorably than robot-assisted interviews. This suggest that expert perceptions of clinical value of the screening interview remain consistent regardless of delivery modality. Given documented concerns about digital mental health integration into clinical practice [42,43] this is a notable finding that contrasts with literature showing implementation barriers and clinician resistance and skepticism towards digital health solutions in mental-health setting [72]. The only significant difference (alignment with other assessment methods) likely reflects integration concerns rather than fundamental clinical utility differences, supporting the concept of digital tools as complementary rather than replacement technologies [12]..

The comparison between humanoid robot and tablet-based approaches highlights important considerations for technology-mediated screening implementation. While the robot condition offered embodied, human-like presence potentially enhancing engagement through social presence and multimodal interaction [23,27], the tablet-based chatbot provides greater accessibility and scalability without specialized hardware requirements. It is worth mentioning that our deterministic, rule-based chatbot approach prioritized clinical safety and transparency over conversational flexibility, ensuring all responses could be validated by healthcare professionals. This design choice reflects current needs to balance technological capability with clinical responsibility in youth mental health applications, particularly given advances in conversational AI showing potential for reducing depression and distress symptoms [19–22,73].

The evaluated approach presented in this paper establishes a foundation for integrating technology-mediated screening into youth mental health services while maintaining clinical standards. Our findings support a complementary model where technology-mediated screening serves as an accessible first step in assessment, informing subsequent clinical decision-making and resource allocation. The demonstrated content validity (H1), successful completion across modalities with equivalent willingness to repeat (H4), and maintained clinical value across most expert-evaluated dimensions (H5) provide compelling evidence for implementation feasibility. The personality factors influencing satisfaction (H3) suggest further opportunities for personalized screening delivery, while the preference patterns (H2) inform user experience design considerations.

The generalizability of this pilot study's methods and findings warrant careful consideration for future definitive trials and broader implementation. Namely, several limitations should be considered when interpreting the results. The study was conducted in a single clinical setting with youth already engaged in mental health treatment, potentially limiting generalizability to broader populations who have not accessed care. The study population's existing engagement with psychiatric treatment may have influenced comfort levels across all modalities, potentially affecting the patterns of satisfaction and willingness predicted in H2. Our focus on immediate feasibility rather than longitudinal outcomes limits conclusions about long-term effectiveness and clinical impact [5,7]. An additional implication of recruiting youth already engaged in treatment is that ongoing interventions may have reduced or stabilized symptom levels by the time of participation [74]. As a result, some clinically relevant difficulties may have been less evident during the screening interview, potentially limiting the range of symptoms that could be detected and leading to conservative estimates of the interview's ability to capture active psychopathology. Future studies should therefore also examine the performance of the screening interview in youth who are not yet in treatment or who present with more acute symptomatology. Furthermore, complete blinding was not feasible due to distinct intervention characteristics. However, structured evaluation protocols help mitigate potential bias in expert evaluations relevant to H3. An important consideration also involves disclosure patterns and social desirability bias. Recent evidence on chatbot-assisted self-assessment demonstrates that individuals are comfortable disclosing sensitive health information to chatbots when anonymity is guaranteed, with previous positive chatbot experiences and favorable attitudes predicting greater information disclosure [26]. While our study did not find technology conditions superior for

satisfaction, previous research suggests youth may be more willing to disclose sensitive information to computer-based systems due to reduced perceived judgment [9,10,27–29]. The comparable clinical value ratings across modalities, however, suggest that any increased disclosure in technology conditions would translate to meaningful clinical information, though this requires further investigation. However, we must note that in study 3, each video was rated by a single expert only, which precluded estimation of interrater reliability. Future studies should ensure that multiple experts rate each interview video so that interrater reliability can be estimated and the robustness of expert-based clinical utility evaluations can be strengthened. Moreover, future research priorities should also include large-scale implementation studies evaluating real-world effectiveness across diverse settings, enhanced technology integration incorporating sophisticated natural language processing and adaptive questioning protocols, and longitudinal validation examining whether technology-mediated screening improves clinical outcomes and reduces access barriers over time [13,14]. The rapid advancement of conversational AI and social robotics technologies suggests significant potential for enhanced screening capabilities [16,22]. Moreover, a within-subject cross-over design might be more sensitive to subtle modality effects but would require counter-balancing and washout periods to reduce carryover effects. If applied in the future the possible residual carryover should be considered as to have impact on interpretation [57].

To sum up, study presented in this paper demonstrates that technology mediated screening interviews represent a feasible and clinically valuable approach for addressing critical barriers in youth mental health assessment. The findings establish that technology-mediated screening maintains clinical and offers a complementary pathway to traditional psychiatrist-led assessments, particularly valuable in settings where access to specialists is limited. For clinical practice, the approach offers transformative potential by expanding early identification capabilities beyond traditional clinical settings, enabling self-administered screening that can triage youth for timely specialist consultation while reducing wait times in overburdened mental health systems. In the current study, technology-mediated interviews did not produce automated reports. Experts evaluated full video recordings, which partially constrains real-world clinical utility. Future work within our group is already underway to derive structured, explainable summaries from recorded interview data and to test how such outputs can be used by clinicians to inform assessment and treatment decisions. Beyond clinical value, the equivalence in willingness to engage across modalities suggests that technology-mediated screening is also feasible to be used by the youth in their daily routine.

Overall, our findings support a complementary model of integration of digital solutions and AI in mental health, where technology-mediated screening serves as an accessible first step in assessment, informing subsequent human-led clinical decision-making and resource allocation [47]. As digital technologies continue advancing and access barriers persist, the approach represents a promising pathway for improving early intervention in youth mental health while preserving the clinical rigor necessary for responsible implementation in vulnerable populations.

## Supporting information

**S1 Table. Summary of the qualitative responses to the first version of the interview questions (Study 1).** (DOCX)

**S2 Table. Diagnoses of participants for the total sample and by experimental conditions (Study 2).** (DOCX)

**S3 Table. Descriptive statistics of and correlations between the central variables – total sample (Study 2).** (DOCX)

**S4 Table. Descriptive statistics of and correlations between the central variables – Psychiatrist condition (Study 2).** (DOCX)

**S5 Table. Descriptive statistics of and correlations between the central variables – Chatbot condition (Study 2).**
(DOCX)

**S6 Table. Descriptive statistics of and correlations between the central variables – Robot condition (Study 2).**
(DOCX)

**S7 Table. Predictors of satisfaction with and willingness to repeat the screening interview in the samples conducting interviews with the help of technology.**
(DOCX)

**S8 Table. Questions for the evaluation of pre-screening interview videos by experts (Study 3).**
(DOCX)

**S9 Table. Comparisons of mean evaluations of variables by four experts (study 3).**
(DOCX)

**S1 Data. Raw data required to replicate the results of your study.**
(ZIP)

**S1 Checklist. CONSORT 2025 checklist.**
(DOCX)

## Acknowledgments

N/A

## Author contributions

**Conceptualization:** Izidor Mlakar, Hojka Gregorič Kumperščak, Sara Močnik, Žan Smogavc, Urška Smrke.

**Data curation:** Izidor Mlakar, Valentino Šafran, Nejc Plohl, Urška Smrke.

**Formal analysis:** Izidor Mlakar, Nejc Plohl, Hojka Gregorič Kumperščak, Urška Smrke.

**Funding acquisition:** Izidor Mlakar, Hojka Gregorič Kumperščak.

**Investigation:** Izidor Mlakar, Valentino Šafran, Nejc Plohl, Hojka Gregorič Kumperščak, Sara Močnik, Žan Smogavc, Urška Smrke.

**Methodology:** Izidor Mlakar, Hojka Gregorič Kumperščak, Urška Smrke.

**Project administration:** Izidor Mlakar.

**Software:** Izidor Mlakar, Valentino Šafran.

**Supervision:** Izidor Mlakar, Hojka Gregorič Kumperščak, Urška Smrke.

**Validation:** Valentino Šafran, Hojka Gregorič Kumperščak, Sara Močnik, Žan Smogavc.

**Visualization:** Izidor Mlakar.

**Writing – original draft:** Izidor Mlakar, Valentino Šafran, Nejc Plohl, Hojka Gregorič Kumperščak, Sara Močnik, Žan Smogavc, Urška Smrke.

**Writing – review & editing:** Izidor Mlakar, Valentino Šafran, Nejc Plohl, Hojka Gregorič Kumperščak, Sara Močnik, Žan Smogavc, Urška Smrke.

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
