## [Decision Letter · Decision Letter 0]

9 Dec 2025

PDIG-D-25-00855Technology-mediated screening interviews for youth mental health: content validation, randomized controlled trial, and expert evaluationPLOS Digital Health Dear Dr. Mlakar, Thank you for submitting your manuscript to PLOS Digital Health. After careful consideration, we feel that it has merit but does not fully meet PLOS Digital Health's publication criteria as it currently stands. Therefore, we invite you to submit a revised version of the manuscript that addresses the points raised during the review process. Please submit your revised manuscript by Feb 07 2026 11:59PM. If you will need more time than this to complete your revisions, please reply to this message or contact the journal office at digitalhealth@plos.org. Please include the following items when submitting your revised manuscript:* A rebuttal letter that responds to each point raised by the editor and reviewer(s). You should upload this letter as a separate file labeled 'Response to Reviewers'. This file does not need to include responses to any formatting updates and technical items listed in the 'Journal Requirements' section below.* A marked-up copy of your manuscript that highlights changes made to the original version. You should upload this as a separate file labeled 'Revised Manuscript with Track Changes'.* An unmarked version of your revised paper without tracked changes. You should upload this as a separate file labeled 'Manuscript'. If you would like to make changes to your financial disclosure, competing interests statement, or data availability statement, please make these updates within the submission form at the time of resubmission. Guidelines for resubmitting your figure files are available below the reviewer comments at the end of this letter. We look forward to receiving your revised manuscript. Kind regards, Phat Huynh

Guest EditorPLOS Digital Health
Phat HuynhGuest EditorPLOS Digital Health
Leo Anthony CeliEditor-in-ChiefPLOS Digital Healthorcid.org/0000-0001-6712-6626
**Journal Requirements:**

1. Please clarify all sources of funding (financial or material support) for your study. List the grants (with grant number) or organizations (with url) that supported your study, including funding received from your institution.

2. State the initials, alongside each funding source, of each author to receive each grant.

3. State what role the funders took in the study. If the funders had no role in your study, please state: “The funders had no role in study design, data collection and analysis, decision to publish, or preparation of the manuscript.”

4. If any authors received a salary from any of your funders, please state which authors and which funders.

2. Please insert an Ethics Statement at the beginning of your Methods section, under a subheading 'Ethics Statement'. It must include:

1) The name(s) of the Institutional Review Board(s) or Ethics Committee(s)

2) The approval number(s), or a statement that approval was granted by the named board(s)

3) (for human participants/donors) - A statement that formal consent was obtained (must state whether verbal/written) OR the reason consent was not obtained (e.g. anonymity). NOTE: If child participants, the statement must declare that formal consent was obtained from the parent/guardian.

3. Your manuscript is missing the following sections: Method. Please ensure these are present, and in the correct order, and that any references to subheadings in your main text are correct. An outline of the required sections can be consulted in our submission guidelines here:

https://journals.plos.org/digitalhealth/s/submission-guidelines#loc-parts-of-a-submission

4. Please provide separate figure files in .tif or .eps format.

5. In the online submission form, you indicated that The datasets used and/or analyzed during the current study are available from the corresponding author upon reasonable request.

The results of this study not publicly available but may be made available to qualified researchers upon reasonable request from the corresponding author.

3. Uploaded as supplementary information.

6. If the reviewer comments include a recommendation to cite specific previously published works, please review and evaluate these publications to determine whether they are relevant and should be cited. There is no requirement to cite these works unless the editor has indicated otherwise.**Reviewers' Comments:** Reviewer's Responses to Questions

**Comments to the Author**

1. Does this manuscript meet PLOS Digital Health’s publication criteria? Is the manuscript technically sound, and do the data support the conclusions? The manuscript must describe methodologically and ethically rigorous research with conclusions that are appropriately drawn based on the data presented.

Reviewer #1: Partly

Reviewer #2: Partly

Reviewer #3: Yes

Reviewer #4: Yes

2. Has the statistical analysis been performed appropriately and rigorously?

Reviewer #1: I don't know

Reviewer #2: Yes

Reviewer #3: Yes

Reviewer #4: Yes

3. Have the authors made all data underlying the findings in their manuscript fully available (please refer to the Data Availability Statement at the start of the manuscript PDF file)?

Reviewer #1: No

Reviewer #2: No

Reviewer #3: Yes

Reviewer #4: No

4. Is the manuscript presented in an intelligible fashion and written in standard English?

Reviewer #1: Yes

Reviewer #2: Yes

Reviewer #3: Yes

Reviewer #4: Yes

5. Review Comments to the Author

Reviewer #1: Title: Technology-mediated screening interviews for youth mental health: content validation, randomized controlled trial, and expert validation.

Summary: There is a critical need to improve the access youth has to mental health professionals. Automated screenings have the potential to democratize access to mental health, and they could reduce the time psychiatrists spend triaging patients. In this study, the feasibility of automated screening interviews is explored. First, a group of expert clinicians assessed the content validity of the structured survey used for the screening. Then, a randomized controlled trial was conducted with pediatric mental health patients where they were assigned to human-led, chatbot, or robot interview. The authors report that although patients preferred the interviews conducted by a human, their willingness to repeat the structured interviews did not change. Lastly, a group of clinicians evaluated the information extracted during the interviews using a structured review (Likert scale type questions). Based on the structured evaluation questions, the authors report the clinicians thought the responses elicited during the chatbot and robot interview to be clinically useful.

Overall Impressions: This study seems well written with much attention to detail. As far as I can assess, the methodology and reporting for Studies 1 and 2 is sound. However, I have some questions regarding Study 3 and a few other minor concerns. I think it would help this manuscript if it were made clearer early on that no AI models were designed or deployed in any of the studies. For a while I was misinterpreting that the chatbot and robot experiments would be conducted with a language model of some sort. Below I’ll break down my concerns by section so that it they are easier to address.

1. Introduction:

1.1. I would recommend intruding concerns regarding AI-safety and trustworthiness during the introduction. Not to discuss them at length but so that the reader knows early on that you’re aware of the barriers in that space, but that this study will not focus on that. For example, the system is not yet at the stage where you’re concerned about how to handle a youth who reports the desire to commit self-harm during an automated screening.

1.2. I think it would be helpful for the reader to understand at this stage why you want to assess the “willingness to repeat” and how that could be interpreted. In my first reading through the manuscript that was a bit unclear for me.

2. Study 1: Content Validity Assessment

2.1. Participant Characteristics

2.1.1. I think it would be more understandable if the authors report the current sample size at each stage of the participant screening process. I was a bit confused with only the percentages.

2.2. Content Validity and Final Interview questions

2.2.1. At this stage of the manuscript CVI and CVR have not been introduced. Expanding the acronyms and giving a short description of what they measure is warranted.

2.2.2. Please review Table 1. There are formatting issues where some columns look odd.

2.2.3. Table 2. Some questions have a star (*) to indicate they were modified, but upon closer inspection, they are the same. One example is questions number 2. Please double check this table.

3. Study 2: Feasibility of AI- and Human-led screening interview – youth perspective

3.1. Why was the study not designed as a within-subjects trial? The willingness of each patient to go through each segment of the trial would be a more direct indicator of “willingness to repeat” than just asking.

4. Study 3: Clinical Utility Assessment

4.1. I have some major concerns about the validity of these findings. How was Inter-rater reliability measured? I would encourage the authors to report a Fleiss’ Kappa. This would allow a reader to understand how strong the conclusions are.

4.2. Did the raters/experts share a set of Psychiatrist, Chatbot, and Robot interviews that could be used a baseline to evaluate their biases?

4.3. How were the survey questions for the expert evaluations designed and validated?

4.4. I think that a really strong demonstration of clinical utility would be if the expert evaluators could deduce the patient’s ICD-10 diagnosis from their interviews. Was this approach considered?

4.5. I feel a better way to frame this study would be as “Perceptions of Clinical Utility” rather than “Demonstrated” clinical utility.

Reviewer #2: STRENGTHS:

This manuscript presents a comprehensive, multi-study validation of technology-mediated screening interviews for youth mental health. This is an area of significant clinical importance given documented shortages in child and adolescent psychiatry. Several aspects of this work merit recognition:

1. Rigorous sequential design: The three-study approach following content validation, user feasibility and expert evaluation represents thorough instrument development methodology. This framework appropriately addresses multiple validity dimensions (content validity, user acceptability, clinical utility).

2. Clinical relevance: The focus on a clinical sample of youth already receiving psychiatric treatment (ages 10-19, n=106) rather than a convenience sample enhances ecological validity. The detailed documentation of participant diagnoses (ICD-10 codes) strengthens the clinical characterization.

3. Theoretical grounding: The authors appropriately situate their work within existing literature on digital mental health interventions, personality predictors of technology acceptance, and barriers to youth mental health care access.

4. Balanced interpretation: The authors appropriately acknowledge that psychiatrist-led interviews achieved higher satisfaction scores while correctly emphasizing that willingness to repeat screening did not differ across modalities.

MAJOR ISSUES:

Study 3 presents a significant methodological concern that could affect the validity of conclusions. The expert evaluation data has a nested/clustered structure: 93 videos were rated by 4 experts with highly unbalanced allocation (Expert 1: 8 videos, Expert 2: 17 videos, Expert 3: 17 videos, Expert 4: 51 videos). This design was analyzed using simple one-way ANOVA, which assumes independence of observations.

This assumption is violated because observations from the same rater are not independent, each rater may have systematic tendencies in leniency/severity or scale interpretation. With one expert contributing 55% of all ratings, any rater-specific bias would disproportionately influence results. The authors should reanalyze Study 3 data using mixed-effects models with rater as a random effect (e.g., Rating ~ Condition + (1|Rater)). This would properly account for the clustering, provide more accurate standard errors and p-values, and allow assessment of between-rater variability. The authors should also report the Intraclass Correlation Coefficient (ICC) to quantify inter-rater reliability and rater agreement.

The robot condition description contains apparent inconsistencies. The authors state the robot was 'remotely controlled by the operator' while also describing autonomous ASR (SPREAD), dialogue management (RASA), and TTS (PLATTOS) systems. The authors should clarify the degree of human operator involvement: Did the operator select responses, monitor an autonomous system with intervention capability, or control non-verbal elements (gestures, movements) while speech was autonomous?

The trial was registered on 16/04/2025 (ISRCTN68006163), but Study 2 data collection occurred from October 2024 to March 2025. This constitutes retrospective registration, which raises concerns about selective outcome reporting. While the authors state "no deviations to the registered protocol," the scientific community cannot verify this without access to the pre-specified analysis plan. The authors should explicitly acknowledge the retrospective registration as a limitation, provide the original statistical analysis plan if available, and clarify whether any outcomes were modified post-hoc.

PLOS Digital Health requires that authors "make all data underlying their findings fully available, without restriction." The current statement ("available upon reasonable request") does not meet this standard. The authors should, if possible, deposit de-identified quantitative data in a public repository (e.g., OSF, Zenodo, Dryad). If there are legitimate restrictions (e.g., ethics committee prohibitions), this must be explicitly stated with the specific restriction and contact information for the body imposing it. The screening interview protocol, questionnaire items, and expert evaluation forms should also be provided as supplementary materials.

The internal consistency for satisfaction with communication in the robot condition (α = .38) is unacceptably low for group comparisons. This threatens the validity of the central finding that psychiatrist-led interviews showed higher communication satisfaction than robot interviews. The authors should report item-level statistics or conduct exploratory factor analysis to understand this pattern, discuss how this reliability issue affects interpretation of between-group differences, and consider whether satisfaction with communication is a valid construct for robot interaction.

MINOR ISSUES

The manuscript uses "AI-assisted" and "AI-mediated" terminology throughout. While this terminology is technically defensible, the system employs Symbolic AI architecture (deterministic state machine for dialogue management) combined with Speech AI components (ASR via SPREAD, TTS via PLATTOS) for natural language interaction, the distinction between this approach and Generative AI (e.g., LLM-based systems) has important implications for generalizability. The authors' design choice to prioritize interpretability and safety through a "white-box" model is appropriate for clinical validation. However, readers should understand that findings may not generalize to Generative AI conversational agents, which would behave fundamentally differently in terms of response flexibility, error patterns, and user interaction dynamics. The authors should add a brief clarification (perhaps in the Methods or Discussion) stating something like: "The system employs a Symbolic AI architecture (deterministic state machine) rather than a Generative AI approach. While this limits conversational flexibility, it provides the interpretability and safety guarantees required for clinical validation. Findings may not generalize to LLM-based conversational agents."

The authors note that several instruments (BIDR-16, G-MISS Communication subscale) were translated into Slovenian using forward-backward translation procedures. However, only English versions of items are presented in the manuscript and supplementary materials. For transparency, reproducibility, and use by researchers in Slovenian-speaking contexts, the authors should provide the administered Slovenian versions of all questionnaire items as supplementary material.

The full regression model in Table 4 includes 12 predictors with n = 101, yielding approximately 8.4 observations per predictor. This is below the conventional 10-15:1 ratio recommended to avoid overfitting and unstable coefficient estimates. Additionally, no multicollinearity diagnostics (VIF values) are reported, despite moderate intercorrelations between several predictors (e.g., Agreeableness and Impression management: r = .53). I recommend the authors to report VIF values to assess multicollinearity, and acknowledge the overfitting risk as a limitation. Consider whether a more parsimonious model might be appropriate.

The manuscript alternates between "pre-screening interview" and "screening interview." The authors should choose one term and use it consistently throughout.

There are quite a lot of typographical and grammatical errors, for example on Page 6: "Traditional clinical interviews, while remaining the gold standard for mental health evaluation, resource-intensive..." missing "are" before "resource-intensive" or on the same page "approximately 20% of adolescents experience a mental health disorder, with half of all lifetime mental health conditions emerging by age 14s" remove "s" from "14s". Please consider re-reading the paper for this type of errors before resubmitting the next version.

Figure 1 and Figure 2 lack descriptive captions explaining what participants saw/experienced.

The exact wording of the 10 screening questions appears in Table 2, but the introduction/contextualization provided to participants (if any) is not described.

The sample is limited to one clinical setting in Slovenia, and all materials were in Slovenian. Cultural and linguistic factors may influence acceptability of technology-mediated mental health assessment. The authors briefly mention this but could expand on implications for cross-cultural implementation.

The discussion would benefit from more specific comparison to prior technology-mediated mental health assessment studies in youth populations. The authors cite relevant literature but do not systematically compare their effect sizes or satisfaction ratings to published benchmarks.

The finding that personality factors (conscientiousness, open-mindedness) predicted satisfaction merits deeper discussion. What are the clinical implications? Should screening implementation be personalized based on personality assessment?

The authors might consider a figure illustrating the sequential study design and participant flow across the three studies.

OVERALL ASSESSMENT AND RECOMMENDATION

This manuscript addresses an important clinical need and presents a methodologically sound validation study of technology-mediated mental health screening for youth. The sequential three-study design is appropriate, and the findings contribute meaningfully to the digital mental health literature. The key finding (that youth show equivalent willingness to engage with technology-mediated screening despite preferring human interaction) has practical significance for implementation. The authors' decision to prioritize clinical safety through a deterministic, interpretable system design is commendable and appropriate for this validation stage. However, several issues require attention before publication (see above), especially the data availability limitation and Study 3 statistical methodology.

Reviewer #3: This paper addresses an important health concern. Worldwide, lack of timely access to specialized mental health assessment limits youth engagement in needed mental health care. The three linked studies are well grounded in existing evidence about the use of artificial intelligence and social robotics and are appropriately designed to fill gaps in the evidence base about assessment of youth.

The paper is clearly written, hypotheses are clearly stated, and the statistical methods are appropriate. The pairwise comparisons of the three assessment conditions are particularly informative. An appropriate statistical power analysis is described.

The major issue with the paper in its current form is that readers cannot meaningfully interpret the results without reading the Methods which come after the Results and Discussion. It appears that this is the journal’s desired format, so it is not clear what the solution is, but readers need the information provided in the Methods section before they read the Results section, particularly with respect to the rating scales and other metrics used throughout the study and the description of the three assessment conditions.

In the initial sections of the paper the term “validation” is used too broadly. In simple terms, “validity” refers to whether the intended construct is actually measured. Only Study 1 is a validation study. The other two studies measure aspects of acceptability, feasibility, and clinical utility. It would help to clearly define these terms and use them consistently. For example, “satisfaction” is often mentioned, but it is not clear whether it is the same as acceptability or whether it has another meaning.

There are some specific points that require clarification:

In the Abstract and the Summary, it is not clear that there are three conditions; it seems that the tablet and robot conditions are somehow combined.

Not all readers will be familiar with the term “2nd Bologna cycle” (page 9).

Some of the potential problems associated with recruiting youth who are in mental health treatment are mentioned as limitations, but it seems that additional potential limitations could be associated with treatment successfully reducing or eliminating symptoms which would limit what could be detected in an assessment.

It is not clear how participants responded in the chatbot and robot conditions—did they respond verbally or did they need to type their responses?

Was there any assessment of interrater reliability in Study 3? If so, it should be briefly described. If not, it should be noted as a limitation.

Several times, it is noted that a potential advantage of technology-based assessment is greater disclosure, especially among youth. However, disclosure is not formally assessed in the study. Is there anything that can be derived from the data on this topic?

The accuracy of the screening is not formally discussed. Since the youth participating in the study were in treatment, it seems that results could be compared to the diagnoses obtained in treatment. Additionally, it would be useful to know if there are differences on any of the study variables associated with diagnosis, or at least, diagnostic categories.

It is not clear whether the chatbot or robot assessments produce any kind of report for clinicians to review and make treatment decisions. Clinical utility seems limited if it is necessary to watch a video recording of the interaction. It may be that this is a goal for future work.

Reviewer #4: This manuscript addresses an important and timely topic in youth mental health, evaluating technology-mediated screening interviews as a complementary tool to traditional psychiatrist-led assessments. The study is well designed, with sequential investigations demonstrating content validity, feasibility, and clinical utility of both chatbot and humanoid robot modalities. Overall, the findings are clear and meaningful, suggesting these tools could expand access to mental health evaluation in underserved settings. I believe the manuscript is of high quality and suitable for publication following minor revisions.

Minor Revision Comments:

Grammar and Formatting:

The manuscript contains minor grammatical and formatting issues, including missing full stops, inconsistent referencing styles, and some repeated information across paragraphs. Careful proofreading is recommended to enhance readability.

Organization of Methods and Results:

Some sections of the manuscript mix methods and results. For clarity, the methods should precede results (page 18). Specifically, participant inclusion criteria for the three studies should be moved from the results section to the methods section (pages 9, 11,14).

Study Design and Statistical Power:

The study employed a between-subject design. The lower statistical power observed could be due to individual variability between groups rather than interview modality. A within-subject design could increase statistical power but might introduce order or carryover effects and participant fatigue. The manuscript should clearly justify the choice of a between-subject design and discuss why participants were not assessed using all three modalities (page 15).

Terminology Regarding AI:

It is unclear whether the study employs true artificial intelligence. The chatbot appears to be rule-based and deterministic, rather than a conversational AI. Terms such as “technology-assisted” or “technology-mediated” may be more accurate than AI (pages 15,21).

Chatbot Development Details:

More details are needed regarding the chatbot platform. Was it developed in-house specifically for this study, or was it based on a publicly available/open-source platform? If in-house, the manuscript should provide programming languages, versions, and development resources used. If open-source, the version should be specified (page 21).

Interview Questions and Standardized Protocols:

The manuscript should clarify whether the interview questions used were based on any standardized protocols or questionnaires commonly used in youth mental health assessments. If not, it would be helpful to explain why these particular questions were chosen and how they differ from established protocols (pages 21,22).

I have attached a draft version of the manuscript with inline comments highlighting specific grammatical, formatting, and structural issues. These should assist the authors in addressing the minor revisions noted above.

6. PLOS authors have the option to publish the peer review history of their article (what does this mean?). If published, this will include your full peer review and any attached files.

**Do you want your identity to be public for this peer review?** For information about this choice, including consent withdrawal, please see our Privacy Policy.

Reviewer #1: No

Reviewer #2: No

Reviewer #3: No

Reviewer #4: No

**Figure resubmission:** While revising your submission, we strongly recommend that you use PLOS’s NAAS tool (https://ngplosjournals.pagemajik.ai/artanalysis) to test your figure files. NAAS can convert your figure files to the TIFF file type and meet basic requirements (such as print size, resolution), or provide you with a report on issues that do not meet our requirements and that NAAS cannot fix.

After uploading your figures to PLOS’s NAAS tool - https://ngplosjournals.pagemajik.ai/artanalysis, NAAS will process the files provided and display the results in the "Uploaded Files" section of the page as the processing is complete. If the uploaded figures meet our requirements (or NAAS is able to fix the files to meet our requirements), the figure will be marked as "fixed" above. If NAAS is unable to fix the files, a red "failed" label will appear above. When NAAS has confirmed that the figure files meet our requirements, please download the file via the download option, and include these NAAS processed figure files when submitting your revised manuscript. **Reproducibility:** To enhance the reproducibility of your results, we recommend that authors of applicable studies deposit laboratory protocols in protocols.io, where a protocol can be assigned its own identifier (DOI) such that it can be cited independently in the future. Additionally, PLOS ONE offers an option to publish peer-reviewed clinical study protocols. Read more information on sharing protocols at https://plos.org/protocols?utm_medium=editorial-email&utm_source=authorletters&utm_campaign=protocols

---

## [Decision Letter · Decision Letter 1]

20 Feb 2026

PDIG-D-25-00855R1Technology-mediated screening interviews for youth mental health: content validation, randomized controlled trial, and expert evaluationPLOS Digital Health Dear Dr. Mlakar, Thank you for submitting your manuscript to PLOS Digital Health. After careful consideration, we feel that it has merit but does not fully meet PLOS Digital Health's publication criteria as it currently stands. Therefore, we invite you to submit a revised version of the manuscript that addresses the points raised during the review process. Please submit your revised manuscript by Mar 22 2026 11:59PM. If you will need more time than this to complete your revisions, please reply to this message or contact the journal office at digitalhealth@plos.org. Please include the following items when submitting your revised manuscript:* A letter that responds to each point raised by the editor and reviewer(s). You should upload this letter as a separate file labeled 'Response to Reviewers'. This file does not need to include responses to any formatting updates and technical items listed in the 'Journal Requirements' section below.* A marked-up copy of your manuscript that highlights changes made to the original version. You should upload this as a separate file labeled 'Revised Manuscript with Track Changes'.* An unmarked version of your revised paper without tracked changes. You should upload this as a separate file labeled 'Manuscript'. If you would like to make changes to your financial disclosure, competing interests statement, or data availability statement, please make these updates within the submission form at the time of resubmission. Guidelines for resubmitting your figure files are available below the reviewer comments at the end of this letter. We look forward to receiving your revised manuscript. Kind regards, Phat Kim Huynh, Ph.D.Guest EditorPLOS Digital Health Phat HuynhGuest EditorPLOS Digital Health Leo Anthony CeliEditor-in-ChiefPLOS Digital Healthorcid.org/0000-0001-6712-6626  **Journal Requirements:** If the reviewer comments include a recommendation to cite specific previously published works, please review and evaluate these publications to determine whether they are relevant and should be cited. There is no requirement to cite these works unless the editor has indicated otherwise.  **Additional Editor Comments (if provided):****Reviewers' Comments:** Reviewer's Responses to Questions

**Comments to the Author**

1. If the authors have adequately addressed your comments raised in a previous round of review and you feel that this manuscript is now acceptable for publication, you may indicate that here to bypass the “Comments to the Author” section, enter your conflict of interest statement in the “Confidential to Editor” section, and submit your "Accept" recommendation.

Reviewer #1: All comments have been addressed

Reviewer #2: All comments have been addressed

Reviewer #3: (No Response)

Reviewer #4: All comments have been addressed

2. Does this manuscript meet PLOS Digital Health’s publication criteria? Is the manuscript technically sound, and do the data support the conclusions? The manuscript must describe methodologically and ethically rigorous research with conclusions that are appropriately drawn based on the data presented.

Reviewer #1: Yes

Reviewer #2: Yes

Reviewer #3: Yes

Reviewer #4: Yes

3. Has the statistical analysis been performed appropriately and rigorously?

Reviewer #1: Yes

Reviewer #2: Yes

Reviewer #3: Yes

Reviewer #4: Yes

4. Have the authors made all data underlying the findings in their manuscript fully available (please refer to the Data Availability Statement at the start of the manuscript PDF file)?

Reviewer #1: Yes

Reviewer #2: Yes

Reviewer #3: Yes

Reviewer #4: Yes

5. Is the manuscript presented in an intelligible fashion and written in standard English?

Reviewer #1: Yes

Reviewer #2: Yes

Reviewer #3: Yes

Reviewer #4: Yes

6. Review Comments to the Author

Reviewer #1: Thank you for thoroughly addressing my comments.

I have one minor suggestion to improve readability. Throughout the text, the term "child and adolescent psychiatrists" is used to refer to "pediatric psychiatrists". I would use the "pediatric" to improve readability. This is nothing major and it's your choice whether to make the changes.

Reviewer #2: (No Response)

Reviewer #3: It does not appear that my comments were addressed in the response. The response mentions only one review and those were not my comments.

Reviewer #4: All major comments have been addressed well and appropriately, and changes made throughout the manuscript.

Minor formatting issues (noted in the previous review) still remain, which might be worth addressing for a good quality publication.

7. PLOS authors have the option to publish the peer review history of their article (what does this mean?). If published, this will include your full peer review and any attached files.

**Do you want your identity to be public for this peer review?** For information about this choice, including consent withdrawal, please see our Privacy Policy.

Reviewer #1: No

Reviewer #2: No

Reviewer #3: No

Reviewer #4: **Yes:** Nathasha Naranpanawa

**Figure resubmission:**  While revising your submission, we strongly recommend that you use PLOS’s NAAS tool (https://ngplosjournals.pagemajik.ai/artanalysis) to test your figure files. NAAS can convert your figure files to the TIFF file type and meet basic requirements (such as print size, resolution), or provide you with a report on issues that do not meet our requirements and that NAAS cannot fix.

After uploading your figures to PLOS’s NAAS tool - https://ngplosjournals.pagemajik.ai/artanalysis, NAAS will process the files provided and display the results in the "Uploaded Files" section of the page as the processing is complete. If the uploaded figures meet our requirements (or NAAS is able to fix the files to meet our requirements), the figure will be marked as "fixed" above. If NAAS is unable to fix the files, a red "failed" label will appear above. When NAAS has confirmed that the figure files meet our requirements, please download the file via the download option, and include these NAAS processed figure files when submitting your revised manuscript. **Reproducibility:** To enhance the reproducibility of your results, we recommend that authors of applicable studies deposit laboratory protocols in protocols.io, where a protocol can be assigned its own identifier (DOI) such that it can be cited independently in the future. Additionally, PLOS ONE offers an option to publish peer-reviewed clinical study protocols. Read more information on sharing protocols at https://plos.org/protocols?utm_medium=editorial-email&utm_source=authorletters&utm_campaign=protocols

---

## [Decision Letter · Decision Letter 2]

18 Mar 2026

Technology-mediated screening interviews for youth mental health: content validation, randomized controlled trial, and expert evaluation

PDIG-D-25-00855R2

Dear dr. Mlakar,

We are pleased to inform you that your manuscript 'Technology-mediated screening interviews for youth mental health: content validation, randomized controlled trial, and expert evaluation' has been provisionally accepted for publication in PLOS Digital Health.

Best regards,

Phat Kim Huynh, Ph.D.

Guest Editor

PLOS Digital Health

**Additional Editor Comments (if provided):**

**Reviewer Comments (if any, and for reference):**

Reviewer's Responses to Questions

**Comments to the Author**

1. If the authors have adequately addressed your comments raised in a previous round of review and you feel that this manuscript is now acceptable for publication, you may indicate that here to bypass the “Comments to the Author” section, enter your conflict of interest statement in the “Confidential to Editor” section, and submit your "Accept" recommendation.

Reviewer #1: All comments have been addressed

Reviewer #2: All comments have been addressed

Reviewer #3: All comments have been addressed

Reviewer #4: All comments have been addressed

2. Does this manuscript meet PLOS Digital Health’s publication criteria? Is the manuscript technically sound, and do the data support the conclusions? The manuscript must describe methodologically and ethically rigorous research with conclusions that are appropriately drawn based on the data presented.

Reviewer #1: Yes

Reviewer #2: Yes

Reviewer #3: (No Response)

Reviewer #4: Yes

3. Has the statistical analysis been performed appropriately and rigorously?

Reviewer #1: N/A

Reviewer #2: Yes

Reviewer #3: (No Response)

Reviewer #4: Yes

4. Have the authors made all data underlying the findings in their manuscript fully available (please refer to the Data Availability Statement at the start of the manuscript PDF file)?

Reviewer #1: No

Reviewer #2: Yes

Reviewer #3: (No Response)

Reviewer #4: Yes

5. Is the manuscript presented in an intelligible fashion and written in standard English?

Reviewer #1: Yes

Reviewer #2: Yes

Reviewer #3: (No Response)

Reviewer #4: Yes

6. Review Comments to the Author

Reviewer #1: (No Response)

Reviewer #2: Thank you for addressing all of our comments. Even though it seems like there was a problem with the transmission of the first round of reviews, the manuscript is in a state acceptable for publication.

The only minor comment which, once addressed, could improve the paper concerns the consistency in terminology regarding the screening interview. It seems like the terms "screening interview" and "pre-screening interview" are used interchangeably throughout the manuscript, most notably, the heading of section 3.2.2 and Table 3 use "pre-screening interview," while Table 4 and the majority of the text revert to "screening interview." Since the authors do describe the tool as a first-step, complementary assessment, "pre-screening" may in fact be the more precise term, but regardless of which label is preferred, we would recommend standardizing it consistently throughout the manuscript.

Reviewer #3: (No Response)

Reviewer #4: All comments have been addressed appropriately.

7. PLOS authors have the option to publish the peer review history of their article (what does this mean?). If published, this will include your full peer review and any attached files.

**Do you want your identity to be public for this peer review?** For information about this choice, including consent withdrawal, please see our Privacy Policy.

Reviewer #1: No

Reviewer #2: No

Reviewer #3: No

Reviewer #4: No
